# Usefulness of Body Fat and Visceral Fat Determined by Bioimpedanciometry versus Body Mass Index and Waist Circumference in Predicting Elevated Values of Different Risk Scales for Non-Alcoholic Fatty Liver Disease

**DOI:** 10.3390/nu16132160

**Published:** 2024-07-07

**Authors:** María Gordito Soler, Ángel Arturo López-González, Daniela Vallejos, Emilio Martínez-Almoyna Rifá, María Teófila Vicente-Herrero, José Ignacio Ramírez-Manent

**Affiliations:** 1Pharmaceutical, 41013 Seville, Seville, Spain; mgorditosoler@hotmail.com; 2Investigation Group ADEMA SALUD, University Institute for Research in Health Sciences (IUNICS), 07010 Palma, Balearic Islands, Spain; d.vallejos@eua.edu.es (D.V.); emilio@mompra.com (E.M.-A.R.); correoteo@gmail.com (M.T.V.-H.); joseignacio.ramirez@ibsalut.es (J.I.R.-M.); 3Faculty of Dentistry, University School ADEMA, 07010 Palma, Balearic Islands, Spain; 4Institut d’Investigació Sanitària de les Illes Balears (IDISBA), Balearic Islands Health Research Institute Foundation, 07010 Palma, Balearic Islands, Spain; 5Balearic Islands Health Service, 07010 Palma, Balearic Islands, Spain; 6Faculty of Medicine, University of the Balearic Islands, 07010 Palma, Balearic Islands, Spain

**Keywords:** non-alcoholic fatty liver disease, visceral fat, body fat, obesity, waist circumference, body mass index

## Abstract

Background: Obesity constitutes a public health problem worldwide and causes non-alcoholic fatty liver disease (MALFD), the leading cause of liver disease in developed countries, which progresses to liver cirrhosis and liver cancer. MAFLD is associated with obesity and can be evaluated by validated formulas to assess MAFLD risk using different parameters such as the body mass index (BMI) and waist circumference (WC). However, these parameters do not accurately measure body fat. As MAFLD is strongly associated with obesity, we hypothesize that measuring body and visceral fat by electrical bioimpedance is an efficient method to predict the risk of MAFLD. The objective of our work was to demonstrate that electrical bioimpedance is a more efficient method than the BMI or WC to predict an elevated risk of MAFLD. Methods: A cross-sectional, descriptive study involving 8590 Spanish workers in the Balearic Islands was carried out. The study’s sample of employees was drawn from those who underwent occupational medicine examinations between January 2019 and December 2020. Five MAFLD risk scales were determined for evaluating very high levels of body fat and visceral fat. The determination of body and visceral fat was performed using bioimpedanciometry. Student’s *t*-test was employed to ascertain the mean and standard deviation of quantitative data. The chi-square test was used to find prevalences for qualitative variables, while ROC curves were used to define the cut-off points for body and visceral fat. The calculations included the area under the curve (AUC), the cut-off points along with their Youden index, sensitivity, and specificity. Correlation and concordance between the various scales were determined using Pearson’s correlation index and Cohen’s kappa, respectively. Results: As both total body fat and visceral fat increase, the risk of MAFLD increases with a statistically significant result (*p* < 0.001), presenting a higher risk in men. The areas under the curve (AUC) of the five scales that assess overweight and obesity to determine the occurrence of high values of the different MAFLD risk scales were very high, most of them exceeding 0.9. These AUC values were higher for visceral and body fat than for the BMI or waist circumference. FLD-high presented the best results in men and women with the AUC at around 0.97, both for visceral fat and total body fat, with a high Youden index in all cases (women body fat = 0.830, visceral fat = 0.892; men body fat = 0.780, visceral fat = 0.881). Conclusions: In our study, all the overweight and obesity scales show a very good association with the scales assessing the risk of MAFLD. These values are higher for visceral and body fat than for waist circumference and the BMI. Both visceral fat and body fat are better associated than the BMI and waist circumference with MAFLD risk scales.

## 1. Introduction

Obesity is a metabolic disorder in which there is an accumulation of excessive fat in the body, which is harmful to health. Nowadays, it is considered a public health problem in both developed and developing countries and is recognized as a 21st century pandemic [1]. The impact of obesity on health is very high, both because of the pathology it presents in itself and because it is a risk factor for multiple diseases. Currently, it is a cause of greater mortality in developed countries than low weight in developing countries [2].

Among the pathologies that obesity promotes are heart disease and stroke [3], diabetes and insulin resistance [4], different types of cancer [3], musculoskeletal disorders [3], and different liver diseases such as non-alcoholic fatty liver disease (MAFLD), non-alcoholic steatohepatitis (MASH), liver cirrhosis, and hepatocellular carcinoma [5,6].

Fatty liver and obesity are closely related. This relationship is due to poor metabolic health that facilitates the development of what can be called fatty liver disease associated with metabolic dysfunction (MAFLD) [7]. MAFLD is currently defined as a chronic liver disease with >5% fat deposition in liver cells in the absence of excessive alcohol intake and other secondary causes [8]. This increase in fat in the liver is responsible for increased insulin resistance, which plays an essential role in the development of MAFLD. The adipocyte behaves like an endocrine organ and produces various cytokines, which include tumor necrosis factor (TNF-α), angiotensinogen, free fatty acids that are mainly responsible for lipotoxicity, and leptin. The latter increases in obese individuals—who produce resistance to it—exert a pro-inflammatory action and the development of MAFLD, and may well play an important role in the progression to liver fibrosis, which, in patients with MAFLD, constitutes the most important risk factor for developing liver cirrhosis and liver cancer [9].

About 20 years ago, MAFLD was strongly associated with obesity, type 2 diabetes mellitus (T2DM), metabolic syndrome (MS), and dyslipidemia [10]. Nowadays, a second type of MAFLD would be related to infectious diseases such as Hepatitis C and HIV or drugs [11]. However, the majority of MAFLD occurs in developed and developing countries due to an unhealthy lifestyle, with an intake of foods rich in carbohydrates and saturated fats (Fast Food), along with a lack of physical exercise, causing obesity [12].

Today, linked to obesity, MAFLD is the leading cause of chronic liver disease in developed countries, with a global prevalence of around 30%, which reaches 90% of patients with morbid obesity [13].

The gold standard for diagnosing any liver pathology is a biopsy; however, it is an aggressive, expensive, painful, and not risk-free test [14].

Liver ultrasound is a non-invasive method and well tolerated by the patient. However, the interpretation of hepatic steatosis is operator-dependent, requires training time for the technician [15], and the classification of steatosis is established subjectively at three levels: mild, moderate, and severe. Further, an experienced sonographer can only detect hepatic steatosis when the fat content in the liver is at least between 2.5 and 20% [16], so approximately 5% of patients with initial steatosis would not be diagnosed [17].

Transient elastography (LSM) is also useful for the diagnosis of MAFLD. However, although its usefulness has been demonstrated in cases of advanced fibrosis in MAFLD, with a sensitivity in these cases of 97% in patients with a BMI < 30 kg/m^2^, it is not useful in incipient or poorly developed cases [18]. It is an operator-dependent technique requiring training time; subcutaneous fat reduces the propagation of ultrasound, thereby affecting the measurements made in people with obesity; and it has the cost of the ultrasound machine [19].

As MAFLD is closely associated with obesity, different formulas have been validated to evaluate the risk of MAFLD. They use different parameters that evaluate obesity, such as the body mass index (BMI) and waist circumference (WC). However, although widely used, these parameters do not accurately measure body fat. The BMI is obtained by dividing weight (in kg) by height (in meters) squared. This formula has several limitations: firstly, it does not provide any information on how body fat is distributed; secondly, when establishing a relationship between height and weight, the BMI is not able to differentiate between muscle, bone, or adipose tissue, so individuals with large muscle mass will obtain a high BMI without having excess adipose tissue [20].

Visceral fat in adults is associated with insulin resistance [21], which plays a prominent role in the development of MAFLD. WC is a valid parameter for indirect measurement of visceral adipose tissue [22]; however, we consider that there are valid economic methods to assess both total body fat and visceral fat. These systems include bioelectrical impedance (BIA), which measures the electrical resistance of the different components of the body by applying a constant low intensity alternating current to them. It is a simple, fast, harmless, low-cost, and affordable technique in medical consultations. This technique is valid for studying body composition in healthy, normal weight individuals and in the study of overweight and moderate obesity. However, it does not seem to have the same usefulness in the study of morbid obesity [23].

It is important to highlight the fact that liver fibrosis can be prevented, reversed, or stabilized if the cause that triggers it is eliminated [24]; and if this is insufficient, immunosuppressive, anti-inflammatory, or antiviral drugs may be considered. New antifibrotic drugs, such as angiotensin inhibitors, are not yet available [25].

As previously mentioned, the most common cause of MAFLD today is unhealthy lifestyle habits, which facilitate the deposition of fat in the liver and are frequently associated with obesity. Modifying lifestyle habits to prevent MAFLD is the basis of preventive and restorative treatment for MAFLD [26]. Therefore, the detection of MAFLD in its early stages is of vital importance for establishing appropriate measures and preventing its evolution to liver fibrosis, liver cirrhosis, and hepatocellular carcinoma.

As MAFLD is strongly associated with obesity, we hypothesize that measuring body and visceral fat by electrical bioimpedance is an efficient method to predict the risk of MAFLD.

The objective of our work was to demonstrate that electrical bioimpedance is a more efficient method than the BMI or WC to predict an elevated risk of MAFLD.

## 2. Material and Methods

### 2.1. Participants

A cross-sectional, descriptive study including 8590 Spanish workers in the Balearic Islands was carried out. The study’s sample of employees was drawn from those who underwent occupational medicine exams between January 2019 and December 2020.

Inclusion criteria:–Individuals in the 18–69 age range;–Consent to take part in the research;–Giving permission for the data to be used for epidemiological research;–Being employed by one of the participating companies in the research and not being temporarily disabled at the time of the study;–A flowchart of the study participants is presented in Figure 1.

### 2.2. Determination of Variables

Following process standardization to prevent interobserver bias, occupational health professionals from the participating firms conducted all measurements, whether anthropometric (height, weight, and waist circumference), analytical, or clinical:–Variables such as age, sex, performing regular physical exercise, physical exercise days per week, and smoking were collected;–Anthropometric and clinical determinations: weight, height, waist and hip circumference, and both systolic and diastolic blood pressure;–Analytical determinations: fasting blood glucose, lipid profile, and hepatic enzymes.

#### 2.2.1. Anthropometric Determinations

Measurements of height (in cm) and weight (in kg) were taken using a SECA 700 scale. The measurements were carried out following the ISAK’s international standards for anthropometric assessment [27].

With the subject standing, feet together, and abdomen relaxed, waist circumference was measured using a tape measure parallel to the floor at the midpoint between the last palpable rib and the iliac crest [28].

Body and visceral fat determination was performed by bioimpedanciometry using a Tanita DC 430MA model. High values of body and visceral fat are considered to be those shown by the bioimpedance scale (from 10 for visceral fat and variable according to age for body fat).

#### 2.2.2. Clinical Determinations

Blood pressure was measured after 10 min of rest, with the subject seated and without crossed legs, using an OMRON-M3 model blood pressure monitor. Three measurements were made at one-minute intervals, and the average of the three was calculated.

#### 2.2.3. Analytical Determinations

The blood sample was taken after a minimum of 12 h of fasting and was then processed in 48 to 72 h. The measurement of triglycerides, total cholesterol, and blood sugar was performed automatically by enzymatic procedures. The dextran sulfate-MgCl_2_ precipitation technique was employed for HDL-cholesterol.

By using the Friedewald formula, which is only reliable when triglycerides do not exceed 400, LDL-cholesterol can be calculated indirectly. The unit of measurement for all analytical parameters is mg/dL.
LDL = Total cholesterol total − HDL-c − triglycerides/5

#### 2.2.4. Risk Scales

The non-alcoholic fatty liver disease and liver fibrosis risk scales listed below were applied:–FLI (fatty liver index) [29] FLI = (e0.953 × loge (triglycerides) + 0.139 × BMI + 0.718 × loge (ggt) + 0.053 × waist circumference − 15.745)/(1 + e0.953 × loge (triglycerides) + 0.139 × BMI + 0.718 × loge (ggt) + 0.053 × waist circumference − 15.745) × 100. High risk is defined as beginning at 60.–Hepatic steatosis index (HSI) [29] HSI = 8 × ALT/AST + BMI + 2 (if type 2 diabetes yes) + 2 (if female). Thirty-six is regarded as high risk.–Zhejian University index (ZJU) [29] ZJU index = BMI + FBG + TG + 3 × ALT/AST +2 (if female). Fasting blood glucose (FBG) was in mmol/L; WC was in cm, triglycerides (TG).–A high-risk situation is defined as 38.–Fatty Liver Disease Index (FLD) [30] FLD= BMI + TG + 3 × (ALT/AST) + 2 × Hyperglycemia (presence= 1; absence = 0)–High risk is defined as beginning at 37.–Lipid accumulation product (LAP) [31]. Men: (waist (cm) − 65) × (triglycerides (mMol)).

Women: (waist (cm) − 58) × (triglycerides (mMol)).

High risk is considered starting from 42.7.

Anyone who had smoked at least one cigarette in the previous month (or its equivalent in other forms of consumption) or who had given up smoking less than a year before was considered a smoker.

The Spanish Society of Epidemiology’s recommendation, based on the 2011 National Classification of Occupations, was used to determine socioeconomic class. Class I comprises managers, directors, and university professionals; Class II includes intermediate vocations and self-employed individuals; and Class III includes manual workers [32].

### 2.3. Statistical Analysis

Student’s *t*-test was employed to ascertain the mean and standard deviation of quantitative data. The chi-square test was used to find prevalences for qualitative variables. ROC curves were used to define the cut-off points for cardiac ages as moderate and high. The calculations included the area under the curve (AUC), the cut-off points along with their Youden index, sensitivity, and specificity. The correlation and concordance between the various scales were determined using Pearson’s correlation index and Cohen’s kappa, respectively. Statistical analysis was performed using SPSS 29.0, with *p* < 0.05 as the recognized threshold for statistical significance.

## 3. Results

The anthropometric and clinical details of the study participants are displayed in Table 1. The analyses comprised a total of 8590 workers (4104 men, 47.8%, and 4486 women, 52.2%). The average age of the sample was marginally higher than 41, with the bulk of participants in the 30- to 49-year-old age range. Labourers were primarily from social class I. In both genders, just over 15% smoked. Of the men and women, 25.9% and 35.1%, respectively, did not exercise regularly.

Table 2 displays the average body and visceral fat values for both sexes based on the results of the various MAFLD risk scales.

The mean body fat values were consistently higher in women and rose as the risk of MAFLD increased. Visceral fat and the MAFLD risk scales showed a similar link but with greater values in men in this instance.

Alongside the rise in the various scales measuring the risk of non-alcoholic fatty liver disease (MAFLD), there was a corresponding rise in the prevalence of extremely high body fat values. This prevalence was typically greater in women. Similar trends were revealed in the prevalence of high visceral fat levels, with men showing the highest prevalence in this instance. Table 3 contains all of the data.

The areas under the curve (AUC) of the five scales that assess overweight and obesity to determine the occurrence of high values of the different MAFLD risk scales were very high, most of them exceeding 0.9. These AUC values were higher for visceral and body fat than for the BMI or waist circumference. In all cases, these AUCs were higher in women. See Figure 2 and Figure 3 and Table 4.

Table 5 presents the cut-off points for all the overweight and obesity scales (with their sensitivity, specificity, and Youden index) to determine elevated values of the different MAFLD risk scales. As already seen for the AUC, the highest Youden index values were found for visceral and body fat, and these values were also higher in women.

Table 6a,b and Table 7a,b present the values of Pearson’s correlation coefficients and Cohen’s kappa indices for both the overweight and obesity scales and the MAFLD risk scales in both sexes. The degree of correlation in both sexes, although especially in women, was generally very high for both scales that assess overweightness and MAFLD.

The degree of concordance, determined with Cohen’s kappa index, shows much lower values for the overweight-obesity scales and higher values for the MAFLD scales, especially FLI with FLD and ZJU and HSI with ZJU.

## 4. Discussion

The ability of electrical bioimpedance to predict the high risk of MAFLD was evaluated in a sample of 8590 Spanish workers of both sexes: 4104 men and 4486 women. All of them belonged to the Autonomous Community of the Balearic Islands (Spain) and were between 18 and 69 years of age.

Regarding the characteristics of the population, it should be noted that they mostly belong to social class I, the majority of whom are between 30 and 49 years of age. The percentage of smokers is the same by sex, around 16%, and the group of women performs less physical exercise than the group of men. It is noteworthy that in our study population, the number of smokers is much lower than the data obtained in the European Health Survey in Spain in 2020, where the smoking population had a percentage of around 20% in women and 30% in men [33]. This could be influenced by the fact that all the participants in the sample come from the Balearic Islands, where the smoking habit is somewhat lower than the national average—approximately 17.2% according to the results of the INE (National Statistics Institute) of Spain in 2022 [34]—which is more in line with our results.

The objective of our study was to demonstrate that electrical bioimpedance is a more efficient method than the BMI or WC in the ability to predict a high risk of MAFLD. To do this, the amount of total body fat and visceral fat determined by electrical bioimpedance was compared against five validated formulas to calculate the risk of MAFLD, of which the BMI or WC are a part.

When we assessed the association between the average values of visceral fat and body fat with the five formulas (FLI, HSI, ZJU, FLD, LAP), we observed that as both total body fat and visceral fat increased, so did the risk of MAFLD, with a statistically significant result in all cases (*p* < 0.001 in all scales). These results coincide with a Spanish study carried out on 219,477 workers, where the relationship between several overweight and obesity scales (including the BMI and different estimators of body and visceral fat) and MAFLD risk scales was evaluated. Their results show that as overweight-obesity increases, the risk of MAFLD also increases in parallel in all the scales used [35]. Our results show that this increased risk is greater in the group of women than in the group of men. The association between increased body fat and MAFLD has already been defined in previous studies, in which an association of up to 80% is established between obesity and MAFLD [36].

When assessing the association between the risk of MAFLD and high body fat values, we found the same results. That is, there was a significant increase in the risk of MAFLD that was more pronounced in the group of women. However, when this association was assessed with high visceral fat values, the increased risk, in this case, was observed to be higher in the group of men. This is consistent with other studies that have found a greater association between visceral fat and the risk of MALFD in men [37]. In men, there is a greater predisposition to the accumulation of visceral fat, with more lipolytic capacity than subcutaneous fat and a greater supply of fatty acids to the liver, which may lead to MALFD [38].

In the analysis of the ROC curves, the AUC of the five scales used to assess overweight and obesity presented a very high value, with all of them showing a result between good and excellent. For women, for all five MAFLD risk scales used, the AUC was higher for both percent total body fat and visceral fat. Even when using the FLD high-risk scale, the AUC gave us a value greater than 0.97, which implies an excellent test result—as such a high result is obtained in very few tests. The same occurred when assessing the percentage of body fat with the FLI high-risk scale, in which the AUC also gave us a value greater than 0.97. Even so, it is worth noting that, for all the risk scales used, the AUC was greater than 0.90 for both the percentage of visceral fat and the percentage of body fat, which implies a very good test result.

In the case of men, the results were similar, although the AUC showed lower values than in women in practically all cases. Even so, we must highlight the highest AUC value when using the FLD high-risk scale for the percentage of total body fat, with a result greater than 0.97, which is excellent. On the other hand, the lowest AUC values were found for the LAP high-risk scales when assessing total body fat, with an AUC of 0.875, and in the HSI high-risk scale when assessing visceral fat, with an AUC of 0.895.

These values, being the lowest AUC obtained, offer a good test result that is very close to very good. Notably, for both risk scales, the AUC presented lower values when related to the BMI or WC.

In the five MALFD risk scales used, the AUC was higher when total body fat or visceral fat was used than when the BMI or WC was used.

The risk of MALFD for all scales was observed to increase more in men than in women as visceral fat increased. In the case of the ROC curves, the AUC for the percentage of visceral fat was higher in women. This tells us that although visceral fat is associated with a higher risk of MALFD in men, the Youden index is higher for women. That is to say, the performance of this test is greater or, in other words, has a greater sensitivity-specificity relationship. It can be seen how it oscillates from the lowest value with a Youden index of 0.708 for high LAP (sensitivity 85.4-specificity 85.4) to the highest value obtained with a high FLD, with a Youden index of 0.892 (sensitivity 94.7-specificity 94.5).

In the Pearson correlation coefficient, a linear correlation can be observed between body fat and visceral fat through the BMI and WC. This strength of correlation is very high, in all cases greater than 0.7 for both sexes, although it shows slightly higher figures in women.

Since MAFLD was not evaluated directly but rather indirectly through MAFLD risk formulas, the Pearson correlation coefficient between the different formulas was also performed to check the validity between them when determining said risk. In all the formulas used, a correlation between high and very high can be seen, with the highest value between ZJU and FLD of 0.996 in women and 0.991 in men, which implies a very high and almost total correlation. The lowest correlation was obtained between HSI and LAP, with a value of 0.678 in women and 0.595 in men. Even so, the results were greater than 0.5 in both cases, which expresses a high or strong correlation.

Likewise, in order to gauge the degree of agreement, Cohen’s Kappa index was evaluated. In the case of comparing the different scales to assess obesity, the greatest degree of agreement was found to have occurred between the BMI and total body fat, and between waist circumference and visceral fat, with good agreement, especially in the group of women. The good agreement between waist circumference and visceral fat confirms that waist circumference is a good indirect measure of the amount of visceral adipose tissue [39].

Regarding Cohen’s Kappa index between the different MAFLD risk formulas, a very good agreement was found between almost all of them except for what corresponds to the LAP. The latter presented values between insignificant and low in both men and women. This could be because the BMI is used in the other four formulas, whereas waist circumference is used in the LAP.

## 5. Strengths and Limitations

Among the strengths of the study, it is worth highlighting the large sample size, almost 9000 persons, and the fact that the determination of body and visceral fat was performed with objective and validated methods, such as bioimpedance measurement.

The main limitation of the study is that MAFLD was not determined by objective methods but by using risk scales, even though these are validated.

## 6. Conclusions

In our study, all the overweight and obesity scales show a very good association with the scales assessing the risk of MAFLD. Nonetheless, these values are higher for visceral and body fat than for waist circumference and the BMI.

Both visceral fat and body fat are better associated than the BMI and waist circumference with MAFLD risk scales.

The use of bioimpedanciometry in primary care and occupational medicine consultations can be very useful in predicting the risk of MAFLD.

## Figures and Tables

**Figure 1 nutrients-16-02160-f001:**
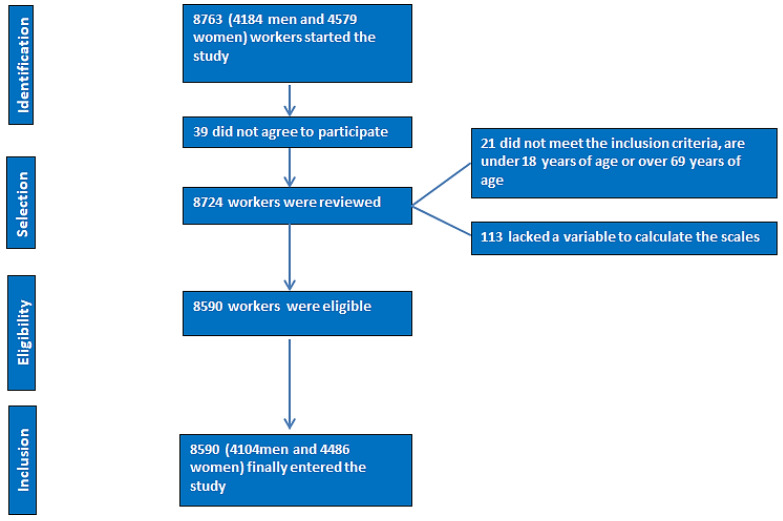
PRISMA flowchart of participants in the study.

**Figure 2 nutrients-16-02160-f002:**
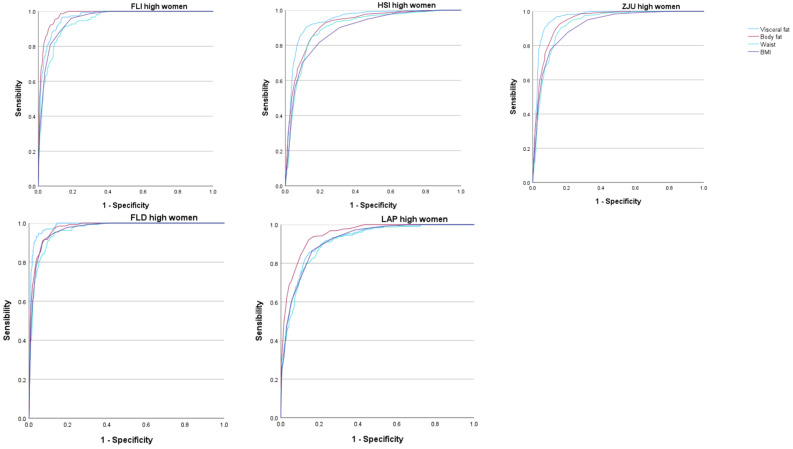
ROC curves in women. FLI Fatty liver index, HSI Hepatic steatosis index, ZJU Zhejian University index, FLD Fatty liver disease index, LAP Lipid accumulation product.

**Figure 3 nutrients-16-02160-f003:**
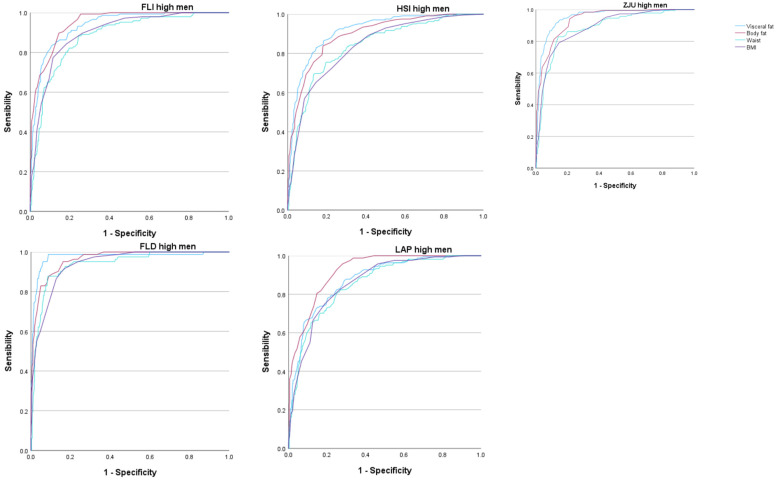
ROC curves in men. FLI Fatty liver index, HSI Hepatic steatosis index, ZJU Zhejian University index, FLD Fatty liver disease index, LAP Lipid accumulation product.

**Table 1 nutrients-16-02160-t001:** Characteristics of study participants.

	Men *n* = 4104	Women *n* = 4486	*p*-Value
	Mean (SD)	Mean (SD)	
Age (years)	41.6 (10.6)	41.5 (10.5)	0.492
Height (cm)	175.8 (7.2)	162.5 (6.1)	<0.001 *
Weight (kg)	81.2 (14.8)	63.9 (13.6)	<0.001 *
Waist circumference (cm)	89.8 (12.5)	77.0 (12.0)	<0.001 *
Hip circumference (cm)	101.8 (8.7)	99.6 (10.9)	<0.001 *
Systolic blood pressure (mmHg)	128.6 (13.3)	117.2 (14.1)	<0.001 *
Diastolic blood pressure (mmHg)	79.9 (10.2)	74.9 (9.9)	<0.001 *
Glycemia (mg/dL)	93.4 (17.8)	88.9 (12.6)	<0.001 *
Total cholesterol (mg/dL)	191.8 (36.0)	189.0 (34.8)	<0.001 *
HDL-cholesterol (mg/dL)	49.2 (11.3)	59.5 (12.8)	<0.001 *
LDL-cholesterol (mg/dL)	124.0 (54.6)	113.8 (30.7)	<0.001 *
Triglycerides (mg/dL)	107.8 (69.4)	81.5 (46.3)	<0.001 *
GGT (UI)	31.5 (30.0)	18.5 (15.9)	<0.001 *
AST (UI)	24.4 (17.3)	18.2 (7.7)	<0.001 *
ALT (UI)	29.3 (34.9)	17.3 (13.4)	<0.001 *
	%	%	*p*-value
18–29 years	15.5	16.8	0.005 *
30–39 years	27.8	25.1	
40–49 years	32.7	34.4	
50–59 years	19.0	19.7	
60–69 years	5.0	4.0	
Social class I	57.1	50.8	<0.001 *
Social class II	20.2	23.8	
Social class III	22.7	25.4	
Non-smokers	84.5	84.2	0.348
Smokers	15.5	15.8	
GGT high	18.6	12.3	<0.001 *
AST high	17.8	11.3	<0.001 *
ALT high	19.5	12.5	<0.001 *
Non physical activity	25.9	35.1	<0.001*
Physical activity 1–3 days/week	27.0	26.5	
Physical activity over 3 days/week	47.1	38.4	

(*) Statistical significance HDL High density lipoprotein. LDL Low density lipoprotein. GGT Gamma-glutamyl transpeptidase. AST Aspartate Aminotransferase. ALT Alanine Aminotransferase UI International unit. SD Standard deviation. (upper normal limits: GGT > 55 men and > 38 women. AST > 33 in both and ALT > 40 men and >35 women).

**Table 2 nutrients-16-02160-t002:** Mean body fat and visceral fat values according to NAFLD risk scale values.

Body Fat		Men			Women	
	*n*	Mean (SD)	*p*-Value	*n*	Mean (SD)	*p*-Value
low FLI	2114	14.7 (5.3)	<0.001 *	3646	26.6 (6.1)	<0.001 *
moderate FLI	930	21.4 (5.0)		478	37.3 (3.8)	
high FLI	1060	27.8 (6.2)		362	42.2 (5.3)	
low HSI	625	11.5 (4.4)	<0.001 *	1320	22.2 (4.8)	<0.001 *
moderate HSI	1015	16.5 (5.0)		1891	28.1 (4.9)	
high HSI	1751	25.5 (6.5)		1275	37.4 (5.7)	
low ZJU	879	11.7 (4.3)	<0.001 *	1589	22.1 (4.6)	<0.001 *
moderate ZJU	1867	17.9 (5.0)		1798	29.4 (4.4)	
high ZJU	1358	27.0 (6.1)		1099	38.4 (5.1)	
low FLD	1293	12.9 (4.7)	<0.001 *	2695	24.4 (5.1)	<0.001 *
moderate FLD	2215	20.4 (5.3)		1477	34.3 (4.3)	
high FLD	596	31.0 (5.7)		314	43.5 (4.8)	
normal LAP	3120	17.7 (6.9)	<0.001 *	3978	27.8 (6.9)	<0.001 *
high LAP	984	27.0 (6.5)		508	40.3 (6.1)	
**Visceral Fat**	** *n* **	**Mean (SD)**	***p*-Value**	** *n* **	**Mean (SD)**	
low FLI	2114	4.8 (2.8)	<0.001 *	3646	3.6 (2.2)	<0.001 *
moderate FLI	930	8.6 (2.9)		478	7.9 (1.8)	
high FLI	1060	13.3 (4.3)		362	10.9 (3.2)	
low HSI	625	3.1 (2.0)	<0.001 *	1320	2.3 (1.3)	<0.001 *
moderate HSI	1015	5.9 (2.9)		1891	4.0 (1.9)	
high HSI	1751	11.4 (4.5)		1275	8.1 (3.3)	
low ZJU	879	3.2 (2.0)	<0.001 *	1589	2.2 (1.3)	<0.001 *
moderate ZJU	1867	6.6 (2.8)		1798	4.4 (1.8)	
high ZJU	1358	12.6 (4.3)		1099	8.6 (3.2)	
low FLD	1293	3.9 (2.4)	<0.001 *	2695	2.9 (1.6)	<0.001 *
moderate FLD	2215	8.1 (3.2)		1477	6.4 (2.3)	
high FLD	596	15.4 (4.2)		314	11.6 (3.0)	
normal LAP	3120	6.4 (3.8)	<0.001 *	3978	4.1 (2.6)	<0.001 *
high LAP	984	12.5 (4.5)		508	9.7 (3.4)	

(*) Statistical significance. FLI Fatty liver index, HSI Hepatic steatosis index, ZJU Zhejian University index, FLD Fatty liver disease index, LAP Lipid accumulation product. SD standard deviation.

**Table 3 nutrients-16-02160-t003:** Prevalence of very high body fat and visceral fat values according to MAFLD risk scales.

Very High Body Fat		Men			Women	
	*n*	%	*p*-Value	*n*	%	*p*-Value
low FLI	2114	0.3	<0.001 *	3646	0.8	<0.001 *
moderate FLI	930	8.6		478	27.6	
high FLI	1060	42.5		362	66.4	
low HSI	625	0	<0.001 *	1320	0	<0.001 *
moderate HSI	1015	0.4		1891	0.4	
high HSI	1751	30.3		1275	31.0	
low ZJU	879	0	<0.001 *	1589	0	<0.001 *
moderate ZJU	1867	1.2		1798	0.3	
high ZJU	1358	38.0		1099	36.2	
low FLD	1293	0	<0.001 *	2695	0	<0.001 *
moderate FLD	2215	4.6		1477	10.3	
high FLD	596	73.3		314	79.5	
normal LAP	3120	5.4	<0.001 *	3978	4.7	<0.001 *
high LAP	984	39.0		508	49.6	
**High Visceral Fat**	** *n* **	**%**	***p*-Value**	** *n* **	**%**	***p*-Value**
low FLI	2114	0	<0.001 *	3646	0.1	<0.001 *
moderate FLI	930	10.2		478	3.2	
high FLI	1060	54.1		362	28.9	
low HSI	625	0	<0.001 *	1320	0	<0.001 *
moderate HSI	1015	2.1		1891	1.7	
high HSI	1751	36.1		1275	9.0	
low ZJU	879	0	<0.001 *	1589	0	<0.001 *
moderate ZJU	1867	1.2		1798	1.1	
high ZJU	1358	47.6		1099	10.4	
low FLD	1293	0	<0.001 *	2695	0	<0.001 *
moderate FLD	2215	10.8		1477	3.3	
high FLD	596	72.0		314	34.8	
normal LAP	3120	6.7	<0.001 *	3978	0.4	<0.001 *
high LAP	984	45.1		508	20.9	

(*) Statistical significance. FLI Fatty liver index, HSI Hepatic steatosis index, ZJU Zhejian University index, FLD Fatty liver disease index, LAP Lipid accumulation product.

**Table 4 nutrients-16-02160-t004:** Areas under the curve for body and visceral fat predict high values of MAFLD risk scales by sex.

*n* = 4486	BMI Women	Waist Women	Body Fat Women	Visceral Fat Women
high FLI	0.952 (0.944–0.960)	0.941 (0.931–0.951)	0.978 (0.974–0.982)	0.965 (0.959–0.972)
high HSI	0.887 (0.876–0.898)	0.900 (0.889–0.910)	0.917 (0.908–0.926)	0.941 (0.933–0.948)
high ZJU	0.914 (0.905–0.924)	0.924 (0.916–0.932)	0.943 (0.936–0.949)	0.964 (0.959–0.969)
high FLD	0.969 (0.962–0.976)	0.963 (0.956–0.971)	0.975 (0.969–0.980)	0.985 (0.981–0.989)
high LAP	0.920 (0.909–0.930)	0.912 (0.901–0.924)	0.950 (0.942–0.958)	0.919 (0.908–0.930)
***n* = 4104**	**BMI Men**	**Waist Men**	**Body Fat Men**	**Visceral Fat Men**
high FLI	0.880 (0.868–0.893)	0.899 (0.888–0.910)	0.936 (0.928–0.944)	0.948 (0.941–0.954)
high HSI	0.838 (0.826–0.851)	0.837 (0.824–0.849)	0.914 (0.905–0.922)	0.895 (0.885–0.905)
high ZJU	0.882 (0.870–0.894)	0.888 (0.877–0.899)	0.952 (0.945–0.958)	0.936 (0.929–0.944)
high FLD	0.937 (0.926–0.948)	0.938 (0.929–0.947)	0.975 (0.966–0.984)	0.963 (0.957–0.970)
high LAP	0.854 (0.841–0.867)	0.860 (0.847–0.872)	0.875 (0.863–0.887)	0.919 (0.911–0.927)

FLI Fatty liver index, HSI Hepatic steatosis index, ZJU Zhejian University index, FLD Fatty liver disease index, LAP Lipid accumulation product. BMI Body mass index.

**Table 5 nutrients-16-02160-t005:** Body and visceral fat cut-off points for predicting high values of MAFLD risk scales by sex.

***n* = 4486**	**Body Fat Women**	**Visceral Fat Women**	**BMI Women**	**Waist Women**
	**Cut-Off-Sens-Specif-Youden**	**Cut-Off-Sens-Specif-Youden**	**Cut-Off-Sens-Specif-Youden**	**Cut-Off-Sens-Specif-Youden**
high FLI	36.5–92.4–92.1–0.845	8.0–89.8–89.5–0.793	28.6–88.2–87.8–0.760	91.0–87.5–87.0–0.745
high HSI	32.8–85.6–84.2–0.698	6.0–89.0–89.0–0.780	25.8–81.7–80.6–0.623	80.0–86.9–86.2–0.731
high ZJU	33.3–88.6–87.1–0.757	6.0–91.8–91.4–0.832	26.6–84.5–82.7–0.672	82.0–91.8–91.4–0.832
high FLD	37.7–91.7–91.3–0.830	8.0–94.7–94.5–0.892	30.4–92.8–90.6–0.814	91.0–90.9–90.2–0.811
high LAP	34.4–87.9–87.6–0.755	6.0––85.4–85.4–0.708	27.1–89.8–81.2–0.710	86.0–83.3–82.4–0.657
***n* = 4104**	**Body Fat Men**	**Visceral Fat Men**	**BMI Men**	**Waist Men**
high FLI	23.1–86.6–86.5–0.731	10.0–86.0–85.5–0.715	27.8–80.9–80.8–0.617	95.0–84.2–81.8–0.660
high HSI	21.3–82.2–81.6–0.638	8.0–83.8–83.5–0.673	26.5–76.3–76.3–0.526	91.0–77.6–72.3–0.499
high ZJU	22.5–85.2–84.2–0.694	9.0–88.8–88.8–0.776	27.4–83.3–82.9–0.662	93.0–82.9–82.5–0.654
high FLD	25.0–89.0–89.0–0.780	11.0–94.2–93.9–0.881	29.9–89.1–86.3–0.754	98.5–89.0–86.2–0.752
high LAP	22.3–83.6–82.0–0.656	9.0–78.7–78.1–0.568	27.0–78.7–77.4–0.561	93.0–76.5–76.4–0.529

FLI Fatty liver index, HSI Hepatic steatosis index, ZJU Zhejian University index, FLD Fatty liver disease index, LAP Lipid accumulation product. BMI Body mass index sens Sensitivity. Specif Specificity.

**Table 6 nutrients-16-02160-t006:** a. Pearson’s correlation. b. Pearson’s correlation.

(a)								
Women		Body fat		Visceral fat		BMI	Waist	
Body fat		1		0.864		0.875	0.834	
Visceral fat				1		0.873	0.846	
BMI						1	0.896	
Waist							1	
**Men**		**Body fat**		**Visceral fat**		**BMI**	**Waist**	
Body fat		1		0.856		0.803	0.789	
Visceral fat				1		0.853	0.842	
BMI						1	0.893	
Waist							1	
**(b)**								
**Women**	**FLI**		**HSI**		**ZJU**	**FLD**		**LAP**
FLI	1		0.865		0.905	0.905		0.853
HSI			1		0.972	0.976		0.678
ZJU					1	0.996		0.755
FLD						1		0.750
LAP								1
**Men**	**FLI**		**HSI**		**ZJU**	**FLD**		**LAP**
FLI	1		0.791		0.898	0.897		0.776
HSI			1		0.938	0.946		0.595
ZJU					1	0.991		0.736
FLD						1		0.728
LAP								1

BMI Body mass index, FLI Fatty liver index, HSI Hepatic steatosis index, ZJU Zhejian University index, FLD Fatty liver disease index, LAP Lipid accumulation product.

**Table 7 nutrients-16-02160-t007:** a. Kappa Cohen indices. b. Kappa Cohen indices.

(a)						
Women	Body fat		Visceral fat		BMI	Waist
Body fat	1		0.396		0.729	0.555
Visceral fat			1		0.343	0.710
BMI					1	0.632
Waist						1
**Men**	**Body fat**		**Visceral fat**		**BMI**	**Waist**
Body fat	1		0.680		0.698	0.629
Visceral fat			1		0.668	0.670
BMI					1	0.729
Waist						1
**(b)**						
**Women**	**FLI**	**HSI**		**ZJU**	**FLD**	**LAP**
FLI	1	0.749		0.902	0.942	0.348
HSI		1		0.918	0.632	0.260
ZJU				1	0.684	0.301
FLD					1	0.267
LAP						1
**Men**	**FLI**	**HSI**		**ZJU**	**FLD**	**LAP**
FLI	1	0.746		0.855	0.806	0.480
HSI		1		0.857	0.608	0.355
ZJU				1	0.719	0.440
FLD					1	0.307
LAP						1

BMI Body mass index, FLI Fatty liver index, HSI Hepatic steatosis index, ZJU Zhejian University index, FLD Fatty liver disease index, LAP Lipid accumulation product.

## Data Availability

Study data are stored in a database that complies with all security measures at the ADEMA-Escuela Universitaria. The Data Protection Delegate is Ángel Arturo López González.

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
