# Peer review of "Usefulness of Body Fat and Visceral Fat Determined by Bioimpedanciometry versus Body Mass Index and Waist Circumference in Predicting Elevated Values of Different Risk Scales for Non-Alcoholic Fatty Liver Disease"

_nutrients, 2024, doi:10.3390/nu16132160_

Round 1
Reviewer 1 Report
Comments and Suggestions for Authors
A. General Comments:
1. The use of serum non-invasive markers as gold standard of NASH is inappropriate and reduces the reliability of results of the study. MRI-PDF or even Fibroscan CAP could have been used instead
2. This is not a review article. Authors should reduce the number of references to less than 40.
3. Please use current terminology of NAFLD (MAFLD) and NASH (MASH) (Gastroenterol Hepatol 2023;19:697).
B. Major Comments:
1. (Abstract): Authors should expand the Results Section by giving the real values and their appropriate statistical parameters and their statistical significance supporting their findings.
2. (Line 72-74): This is not true. The association of NAFLD with the metabolic syndrome and dyslipidemia was recognized more than 20 years ago (see: Diabetes Care 1991:14:173, Clin Nutr 1999:18:353, Diabetes 2001;50:1844, Best Pract Res Clin Gastroenterol 2002;16:709). Only the terminology has been changed recently.
3. (Table 1): What are the upper normal limits (UNL) of AST, ALT and GGT for men and women? It is suggested to replace these variables with the respective UNL and then make the statistical comparisons.
4. (Table 2): Please explain the abbreviations at the bottom of the table.
5. (Line 218): The sentence "Men had more negative ....." must change. Please explain the meaning of "negative" or omit the sentence altogether.
6. (Line 365): The cost of bioimpedanciometry apparatus (Tanita DC 430MA model) is €3350) may be a problem. It is good to know that bioimpedanciometry is a better method to estimate body and splanchnic fat than BMI and abdominal circumference, but the later are much more inexpensive.
C. Minor Comments:
1. A few typing errors must be corrected.
2. (Line 119): The authors should note that antifibrotic agents for treatment of advanced NASH are not available in the clinical practice as yet.
Author Response
Dear reviewer,
First of all, thank you for your work and all your recommendations.
To facilitate your review, we have written the modifications in red in the article.
- General Comments:
- The use of serum non-invasive markers as gold standard of NASH is inappropriate and reduces the reliability of results of the study. MRI-PDF or even Fibroscan CAP could have been used instead.
Thank you very much for your recommendation.
We completely agree with you that the certain diagnosis of NASH is obtained by imaging techniques (ultrasound, MRI) or by histology. In this study, these techniques were not performed due to the impossibility of assuming the cost of almost 9000 tests, and it was decided to use different NASH risk scales, which are validated, due to their low cost and taking into account that one of the The objectives of the study were to easily and efficiently detect people at high risk of NASH in order to act on them preventively.
- This is not a review article. Authors should reduce the number of references to less than 40.
Following your recommendations, we have reduced our references to less than 40.
- Please use current terminology of NAFLD (MAFLD) and NASH (MASH) (Gastroenterol Hepatol 2023;19:697).
Following your recommendations, we have proceeded to make all the modifications to the terminology in the manuscript as you recommended.
- Major Comments:
- (Abstract): Authors should expand the Results Section by giving the real values and their appropriate statistical parameters and their statistical significance supporting their findings.
Following their advice, we have proceeded to expand the results section, by providing the results with their appropriate statistical parameters and their statistical significance supporting our findings. Thank you very much
- (Line 72-74): This is not true. The association of NAFLD with the metabolic syndrome and dyslipidemia was recognized more than 20 years ago (see: Diabetes Care 1991:14:173, Clin Nutr 1999:18:353, Diabetes 2001;50:1844, Best Pract Res Clin Gastroenterol 2002;16:709). Only the terminology has been changed recently.
Thank you very much for your observation. We have proceeded to rectify these lines according to his advice.
- (Table 1): What are the upper normal limits (UNL) of AST, ALT and GGT for men and women? It is suggested to replace these variables with the respective UNL and then make the statistical comparisons.
Following their recommendation, in Table 1, we have recorded the upper normal limits (UNL) of AST, ALT and GGT for men and women, making the statistical comparison between sexes.
- (Table 2): Please explain the abbreviations at the bottom of the table.
Following your recommendations, we have proceeded to detail the abbreviations at the end of all tables.
- (Line 218): The sentence "Men had more negative ....." must change. Please explain the meaning of "negative" or omit the sentence altogether.
You are right, the phrase is not clear enough, so we have proceeded to delete it. Thank you so much.
- (Line 365): The cost of bioimpedanciometry apparatus (Tanita DC 430MA model) is €3350) may be a problem. It is good to know that bioimpedanciometry is a better method to estimate body and splanchnic fat than BMI and abdominal circumference, but the later are much more inexpensive.
We are aware of the cost of the Tanita DC 430MA, however, we consider that it can be cost-efficient in the National Health system of our country. Visceral obesity and hepatic steatosis constitute a growing problem in the entire world population, and with a great economic cost for health systems. In Spain, investments are being made in purchasing ultrasound machines in the different Autonomous Communities for Primary Care Health Centers due to their high efficiency. The price of the Tanita DC 430MA is much lower than that of an ultrasound machine. The fact that body or visceral fat determined by bioimpedance is not part of the NASH formulas, unlike BMI or waist, and that these variables (visceral fat and body fat) correlate better with the high risk of presenting NASH is very interesting and is one of the main contributions of the study. Therefore, we consider that the acquisition of a bioimpedanciometry device by a Primary Care Health Center would be cost-effective, in order to detect those people with a high risk of suffering from non-alcoholic hepatic steatosis.
Thank you very much for your warning.
- Minor Comments:
- A few typing errors must be corrected.
The text has been reviewed and corrected by Meryl Jons, professional translator of medical manuscripts, Academia WYN in Mallorca. We trust that all defects have been corrected. Thank you so much.
- (Line 119): The authors should note that antifibrotic agents for treatment of advanced NASH are not available in the clinical practice as yet.
Thank you very much for your observation. We have noted in the manuscript that antifibrotic agents for the treatment of advanced NASH are not yet available in clinical practice.
Thank you very much for your suggestions. We have proceeded to answer all of them and we trust that they will adequately respond to your questions.

Reviewer 2 Report
Comments and Suggestions for Authors
This manuscript assesses the potential usefulness of bioimpedance-mediated body fat measurement in predicting the risk of non-alcoholic fatty liver disease, mainly focusing on a predictability comparison between the total body and visceral fat content with the existing body composition risk factors BMI and waist circumference. The paper is strong in analyzing a large population dataset. The data processing and statistical analysis are mainly valid. However, there are obvious caveats in study design, data presentation, and result interpretation. It is not suitable for publication at its current stage.
Major concerns:
1. The study design is invalid in reaching the author's conclusion. Because all the existing NAFLD indexes have a component of BMI and WC, the correlation of body fat with the index scores actually reflects the correlation between body fat and BMI and WC. To address the author's research question for the predictability of either set of body values, an independent method to assess NAFLD conditions, such as imaging or histology, is needed.
2. For All tables, please use the * symbol to indicate the significantly changed values.
Author Response
Dear reviewer,
First of all, thank you for your work and all your recommendations.
To facilitate your review, we have written the modifications in red in the article.
This manuscript assesses the potential usefulness of bioimpedance-mediated body fat measurement in predicting the risk of non-alcoholic fatty liver disease, mainly focusing on a predictability comparison between the total body and visceral fat content with the existing body composition risk factors BMI and waist circumference. The paper is strong in analyzing a large population dataset. The data processing and statistical analysis are mainly valid. However, there are obvious caveats in study design, data presentation, and result interpretation. It is not suitable for publication at its current stage.
Major concerns:
- The study design is invalid in reaching the author's conclusion. Because all the existing NAFLD indexes have a component of BMI and WC, the correlation of body fat with the index scores actually reflects the correlation between body fat and BMI and WC. To address the author's research question for the predictability of either set of body values, an independent method to assess NAFLD conditions, such as imaging or histology, is needed.
Thank you very much for your observation. We completely agree with you that the certain diagnosis of NASH is obtained by imaging techniques (ultrasound, MRI) or by histology. In this study, these techniques were not performed due to the impossibility of assuming the cost of almost 9000 tests and it was decided to use different NASH risk scales, which are validated, due to their low cost and highlighting that one of the objectives of the study was to determine people with a high risk of NASH in order to act on them preventively.
The interesting thing about our study is that body and visceral fat, which do not appear in the NASH formulas like BMI or waist do, show a better correlation with NASH. Which would allow easier and greater detection of people at high risk of NASH in Primary Care and Occupational Medicine offices. In order to act early in a preventive manner in these patients.
- For All tables, please use the * symbol to indicate the significantly changed values.
Thank you very much for your observation, we have proceeded to add the * symbol in all tables in those values that have changed significantly.
Thank you very much for your suggestions. We have proceeded to answer all of them and we trust that they will adequately respond to your questions.
